# Western Corn Rootworm (*Diabrotica virgifera virgifera* LeConte) in Europe: Current Status and Sustainable Pest Management

**DOI:** 10.3390/insects12030195

**Published:** 2021-02-25

**Authors:** Renata Bažok, Darija Lemić, Francesca Chiarini, Lorenzo Furlan

**Affiliations:** 1Department for Agricultural Zoology, Faculty of Agriculture, University of Zagreb, Svetosimunska 25, 10000 Zagreb, Croatia; rbazok@agr.hr (R.B.); dlemic@agr.hr (D.L.); 2Veneto Agricoltura, Agricultural Research Department, 35020 Legnaro, PD, Italy; francesca.chiarini@venetoagricoltura.org

**Keywords:** IPM, WCR distribution, WCR damage, crop-rotation-based solutions, agronomic alternative, eradication, containment

## Abstract

**Simple Summary:**

*Diabrotica virgifera virgifera*, also known as western corn rootworm (WCR), is a maize-specific pest that has been a serious threat in Europe since the mid-1990s. Between 1995 and 2010, European countries were involved in international projects to plan pest control strategies. However, since 2011, collaborative efforts have declined and the overview of knowledge on WCR is in great need of updating. Therefore, a review of scientific papers published between 2008 and 2020, in addition to direct interviews with experts responsible for WCR management in several European countries, was conducted to (1) summarize the research conducted over the last 12 years and (2) describe the current WCR distribution and population in the EU, and the management strategies implemented. A considerable amount of new knowledge has been gained over the last 12 years, which has contributed to the development of pest management strategies applicable in EU agricultural systems. There is no EU country reporting economic damage on a large scale. In many countries, solutions based on crop rotation are regularly implemented, avoiding insecticide use. Therefore, WCR has not become as serious a pest as was expected when it was discovered in much of Europe.

**Abstract:**

Western corn rootworm (WCR), or *Diabrotica virgifera virgifera* LeConte, became a very serious quarantine maize pest in Europe in the mid-1990s. Between 1995 and 2010, European countries were involved in international projects to share information and plan common research for integrated pest management (IPM) implementation. Since 2011, however, common efforts have declined, and an overview of WCR population spread, density, and research is in serious need of update. Therefore, we retained that it was necessary to (1) summarize the research activities carried out in the last 12 years in various countries and the research topics addressed, and analyze how these activities have contributed to IPM for WCR and (2) present the current distribution of WCR in the EU and analyze the current population levels in different European countries, focusing on different management strategies. A review of scientific papers published from 2008 to 2020, in addition to direct interviews with experts in charge of WCR management in a range of European countries, was conducted. Over the past 12 years, scientists in Europe have continued their research activities to investigate various aspects of WCR management by implementing several approaches to WCR control. A considerable amount of new knowledge has been produced, contributing to the development of pest management strategies applicable in EU farming systems. Among the 10 EU countries analyzed, there is no country reporting economic damage on a large scale. Thanks to intensive research leading to specific agricultural practices and the EU Common Agricultural Policy, there are crop-rotation-based solutions that can adequately control this pest avoiding insecticide use.

## 1. Introduction

The most severe maize pest in North America, the western corn rootworm (*D. virgifera virgifera* LeConte) (WCR), known as the billion-dollar beetle [1], was first discovered in Europe in 1992 near Belgrade, Serbia [2]. 

Immediately after the news of this pest arrival had spread among scientists, it became clear that international action against the WCR would be necessary. The International Working Group on *Ostrinia* and other Maize Pests (IWGO), which was established in the mid-20th century as part of the International Organization for Biological Control (IOBC) to study the European corn borer (ECB) and other corn pests, included the WCR in its program activities. Since 1995, IWGO, in collaboration with the European and Mediterranean Plant Protection Organization (EPPO) and Food and Agriculture Organization (FAO) of the United Nations (UN), has organized annual meetings to share new information among scientists on pest distribution and the damage caused. WCR is still on the agenda on regular biannual IWGO meetings. This made the WCR the only pest in the world to be monitored using the same method in most countries, and its spread was determined in detail every year. The threatened Eastern European countries were particularly active in this sense. The monitoring of adult WCR by European countries enabled the rapid detection and determination of the spread of this invasive pest species since its first observation [3]. Permanent monitoring stations were established by each network partner. These stations enabled the measurement of population fluctuations over the years.

Since the 1980s, there have been three phases of WCR invasion in Europe [4]—the first phase was from accidental introduction until the pest was first identified in the maize field in Serbia. Accidental introduction in Europe took place, according to Szalai et al. [5], between 1979 and 1984, i.e., 8 to 13 years before this species was discovered and started to damage maize fields. The second phase was the spread and establishment of WCR in the Eastern European countries that surround Serbia (i.e., Hungary, Croatia, Romania, Bulgaria, and Bosnia and Herzegovina). The third phase of invasion (2001–2018) was a dispersal phase in which WCR spread across most European countries (EPPO) [6].

According to WCR population genetics studies by Miller et al. [7] and Ciosi et al. [8], WCR was introduced into Serbia, with the population source probably being Pennsylvania. From Serbia, the pest spread naturally to most of the countries of central and south-eastern Europe, in addition to the Italian region of Friuli. Later on, there were four other introductions of WCR into several other European countries: Italy (Lombardy region), France (Alsace region), France (Paris region), and the United Kingdom [8].

A significant bulk of new WCR knowledge has been created by Europe’s scientific communities in the past 25 years. At the very beginning, European research focused on WCR monitoring and spread [3,9,10,11,12,13,14,15,16,17,18], predictions of its further spread and damage [19,20,21,22,23,24,25,26], tools for monitoring [27,28], ecology [13,29,30,31,32,33,34,35], and damage [36], and on control methods [37] and tools, including biological control [38,39,40]. 

This paper aims to analyze what has happened since the end of FAO and EU monitoring and research projects in different areas of the EU, and what the current WCR situation is in Europe in terms of research, spread, population levels, damage, and control measures. Our aims were to (1) summarize the research activities carried out in the last 12 years in various countries and the research topics addressed, and analyze how these activities have contributed to IPM for WCR and (2) present the current distribution of WCR in the EU and analyze the current population levels in different European countries, focusing on different management strategies.

## 2. History of WCR Management in Europe

The first international project “Development and Implementation of Containment and Control of the Western Corn Rootworm in Europe” was implemented from 1997 to 2000. It was founded by FAO. In 2003, a new FAO project “Integrated Pest Management for Western Corn Rootworm in Central and Eastern Europe (GTFS/RER/017/ITA)” was launched in the seven most endangered countries (Hungary, Croatia, Serbia, Bosnia and Herzegovina, Slovakia, Romania, and Bulgaria) and implemented until 2008. Participatory research activities, field studies, and field-training sessions implemented in each country have demonstrated successful management approaches for controlling WCR in a range of agroecological and socioeconomic conditions. The introduction of farmer field schools (FFSs) and student field schools (SFSs) provided an innovative model of working with farmers, and for collaboration among farmers. The original focus on WCR risk management has also widened and led to a better understanding of local agrobiodiversity. The involvement of additional institutions (secondary schools, local and regional administrations) increased WCR awareness and led to new approaches to agricultural extension. Regional networking contributed to reaching a common understanding in all participating countries, from training activities to WCR monitoring and research [41]. 

However, only countries from east and central Europe were involved in the project. In many countries, the project was carried out by quarantine officials and not by scientists. The scientists involved very often had weak links with scientists from more developed EU countries and with better research infrastructure. Due to the quarantine status and weak research infrastructure, no laboratory colonies were available and all research activities were conducted in field conditions depending on the fluctuating WCR population, from very low to very high. As a result, the research results were not widely disseminated in the scientific community and had a limited impact on further research within the EU.

The first EU-funded scientific project on WCR, “Threat to European Maize Production by Invasive Quarantine Pest, Western Corn Rootworm (*Diabrotica virgifera virgifera*): A New Sustainable Crop Management Approach”, conducted from 2000 to 2003, focused on eradication and containment measures. As a result, measures including crop management, plant-insect interactions, natural enemy assessment, risk management, and biotechnological control were investigated and developed [42]. The project results contributed to the creation of an EU strategy to contain and/or eradicate the pest, and in 2003, the Commission of the European Union issued emergency measures (Commission Decision 2003/766/EC of 24 October 2003 on emergency measures to prevent the spread of *Diabrotica virgifera virgifera* LeConte within the community) [43]. In 2006, the 2003 regulation was supplemented by Decision 2006/564/EC [44], which introduced additional requirements for the containment of WCR in the infested zones to limit the further spread of the pest. European Commission (EC) Recommendation 2006/565/EC [45] made it possible to switch from an eradication policy to a containment policy. However, from 2000 to 2009, WCR spread extensively over non-EU territory, and over some EU countries [4].

As the result of the second EU-funded project “Harmonise the Strategies for Fighting *Diabrotica virgifera virgifera*” implemented between 2006 and 2008 [46], several control strategies for WCR management were explored, including biological control, utilization of plant resistance traits, plus the adaptation of biotechnological approaches and cultural techniques. All of the explored measures had to carry a minimum impact on biodiversity and the environment. Additionally, a database was constructed containing all available literature on WCR ecology and current research activities, and a comprehensive review was written of past research focusing on WCR biology. Experts in maize agriculture and WCR ecology were brought together to develop biological WCR control strategies that could integrate with established control options used against other maize pests. Researchers also looked into the possibility of enhancing and maintaining various natural WCR enemies to reduce pest outbreaks.

Additional research has also been financed by national sources. Among the many national programs, the most comprehensive was the German *Diabrotica* research program financed by Germany’s Ministry of Food, Agriculture, and Consumer Protection and implemented from 2008 to 2012. This program consisted of 11 research activities carried out at the federal level and 12 research activities carried out locally, with the region of Bavaria being the most endangered region in Germany [47].

In the past 10 years, WCR has not been considered a new pest in many EU countries because it became a regular part of entomofauna. Moreover, research and/or monitoring activities were organized at a national (or even local) level. WCR was removed from the EU quarantine list in 2014 [48]. However, it remains on the A1 or A2 list of some European countries that are non-EU members, such as Azerbaijan (on A1 list since 2007), Georgia (on A1 list since 2018), Moldova (on A1 list since 2006), Russia (on A1 list since 2014), Turkey (on A1 list since 2016), and Ukraine (on A2 list since 2019). 

In the EU, solutions to manage WCR damage in maize must comply with current legislation requiring the implementation of the principles of integrated pest management (IPM), as described in Annex III of Directive 2009/128/EC [49]. The first IPM step is prevention, i.e., the implementation of a set of agronomic measures such as crop rotation (the first measure listed in Annex III) and, where appropriate, the use of resistant/tolerant varieties, which create the conditions for reducing the risk of pest outbreaks and thus the need for plant protection measures. 

Because the most effective strategy against WCR is rotation [3,50,51], its implementation is made mandatory by the aforementioned legislation, but this may be a problem for livestock farms that have to maintain forage production at the best level in terms of yield and quality. 

## 3. Research Activities on WCR in Europe and Topics Investigated 

Google Scholar literature review was queried using the following keyword combinations: “western corn rootworm in Europe,” “*Diabrotica virgifera virgifera* in Europe,” etc. The search was limited to scientific articles or communications published in English, Croatian, Serbian, and German from 2008 to 2020 and researches conducted on the European area. We aimed to create an overview of all European research groups, a list of research topics, and a reference set of published articles. In Figure 1, the main research areas and number of published papers by each area have been presented. 

Our review of research activities was composed of 187 relevant references covering seven WCR research areas in Europe over the last 12 years. All research areas are divided into sub-themes describing the focus of the research conducted. The main findings of each research area have been listed. The country of affiliation of the authors in the overview of research activities is also listed. The reference list of papers relevant to each defined subtopic is given in Table 1, and a brief description of all research areas is presented here with the main methods and results.

We organized the evaluation of WCR research areas in Europe into seven main categories, depending on the primary type of interest, namely, (1) monitoring and density estimation; (2) ecology; (3) morphology and physiology; (4) trophic interactions; (5) pest control; (6) population genetics; and (7) systems modeling.

From the collected set of references, we first extracted those specifically dealing with “monitoring” procedures in Europe. Thus, we included all papers describing the first WCR occurrence, studies of population levels during the first WCR invasion process, and recorded movements of WCR individuals at short and long distances. The monitoring-based work described in detail the three phases of the WCR invasion process—introduction, establishment and spread, and the influences of various biotic and abiotic factors on WCR (e.g., weather, host plants). A major sub-theme of this research area is the description of different monitoring techniques and procedures that are integrated with the standard monitoring process in Europe. It was important to include “density estimation” because it is a research area closely related to monitoring, as the listed papers deal with different methods, measurements, and estimations of WCR population density, and with climatic conditions and climate change impacts on WCR in monitored areas in Europe. We divided this common research area into four main categories, depending on the type of monitoring and the tool used to describe population density, namely, (1) initial occurrence and spread; (2) population-level; (3) monitoring methods/techniques and designs; and (4) area-wide monitoring. 

International collaborations in the field of WCR “ecology” over the past 12 years have addressed climate change and its influence on expanding areas of WCR invasion. Moreover, dozens of studies have been conducted on various WCR host opportunities, ground preferences, and movements between and within areas. Particular attention has been paid to larval behavior in response to root escapes and to the identification of various attractants that provide communication channels for reproduction and feeding, respectively. 

“Morphology and Physiology” of WCR is the third research area and is represented by three sub-themes—(1) dimorphism, (2) wing morphology, and (3) enzyme activity. Researchers were concerned with specific changes in WCR explained in the context of natural selection, flight maneuverability, invasion process, resistance evolution, etc. 

“Trophic interactions” were the subject of a small research area dealing with (1) WCR-maize-root interactions, (2) plant signals, and (3) WCR vectoring abilities. The main focus was on exploring ways to enhance biological control by manipulating the production of and responsiveness to plant signals. Smaller groups addressed bacterial and fungal community shifts in response to larval feeding and the identification of WCR larvae as potential disease vectors on maize plants. 

Research addressing the “population genetics” of WCR in Europe focused mainly on the genetic basis of WCR dispersal and the temporal and spatial genetic monitoring that allows a deeper understanding of the changes that WCR populations have undergone as a result of the replacement of their original habitat in the USA with a new one in Europe. 

The largest research area in Europe was “pest control” as a topic of interest to researchers from all WCR-infested countries. Special attention was given to risk assessment and prediction of WCR in all maize-growing areas in Europe. Crop rotation was explored as the most effective control measure against WCR. In contrast to America, few scientific groups in Europe have focused on WCR resistance. Furthermore, conventional genetic research on finding tolerant maize hybrids through intensive breeding programs for native resistance (e.g., extended root systems, ion concentrations in roots) has been very limited. Some papers have identified chemical insecticide control (either in-furrow microgranules or seed coating) as a factor that reduces root and yield damage. Even trials with the most effective insecticides show that pesticide protection is partial; they also note that when significantly lower than untreated plots, root damage in treated plots remains appreciable. No experiment made clear whether WCR root damage, usually still present in treated plots, leaded to yield reductions or whether it leaded to a yield comparable to that of plants with uninfested root systems in maize rotation fields. When WCR populations exceed thresholds, first-year maize fields in the rotation are the only ones suffering negligible root damage. Various non-chemical control measures were explored, with entomopathogenic fungi and nematodes identified as having a high potential to reduce WCR larvae in most European soils. All this led to commercial mass production of these environmentally safe control agents. Moreover, biopesticides and natural WCR enemies were identified as useful elements for a strategic approach to WCR pest control. 

We found that over the past 12 years, many groups from Europe have been involved in predicting and assessing the further spread of WCR, its evolution, and the achievement of thresholds. This has led to a mechanistic understanding of the maize–WCR system and the development of remote sensing models that identify WCR larval damage, aspects of the invasion, growth rate, hatching prediction, effects of climate change, and crop rotation on spread and occurrence, etc. In European countries, the need for knowledge transfer, training for farmers, and the need for regulations for the introduction of alternative pest control options have been identified. Long-term and proactive coordination is required for the implementation of collective WCR control measures that meet the needs of individual farmers. Through farmer field schools, farmers were educated on WCR risk assessment and integrated pest management, resulting in successful WCR control in Europe.

## 4. Current Status of the Pest in Europe 

### 4.1. Pest Distribution

To determine the current distribution of the pest in Europe, we consulted the EPPO Global Database (2020) [48] and the available literature sources found by searching Web of Science, SCOPUS, CAB direct, and Google Scholar databases. Based on the collected information, we created a map of WCR distribution in Europe. 

According to the EPPO Global Database [48] and other literature data [52,58,62,69,243], WCR is currently distributed across 21 European countries (Figure 2). In the United Kingdom and the Netherlands, WCR has been successfully eradicated, and in Belgium, the pest no longer exists even though the eradication has not been carried out. In Denmark, Estonia, and Spain, the absence of the pest is confirmed by surveys. In Finland, no pest has been recorded. The degree of WCR distribution varies in each European country—in some, the pest is widespread (the dark red color on the map); in others, it is distributed across a limited area, and this limited area of pest distribution corresponds to the area suitable for maize cultivation (e.g., Austria, Croatia, Italy, Montenegro). In several countries, including Germany and France, the restricted area is limited to restricted regions (orange color on the map). Here, the WCR population is monitored and suppressed by containment measures after the first introduction. Several countries are in the process of eradicating the pest (e.g., Switzerland).

### 4.2. WCR Population Level, Damage, and Management Practices

WCR population level, damage, and management practices were analyzed in 10 European countries: Austria, Croatia, France, Germany, Hungary, Italy, Serbia, Slovenia, Switzerland, and Romania. Based on the EPPO Global Database [48], the selected countries were divided into the following three categories according to their pest distribution status:

Category I. Countries infested with WCR either in the whole maize-growing area or in the whole territory—Austria, Croatia, Hungary, Italy, Serbia, Slovenia, and Romania reported damage in the 2000s.

Category II. Countries partially infested—Germany and France.

Category III. Countries where containment measures have been moderately successful—Switzerland.

To obtain information on WCR population levels, damage, and control measures in various EU countries, we reviewed available literature sources regarding the period between 2008 and 2020. We used the overview explained in Section 2. All abstracts of publications were screened, and literature references were chosen for their relevance to the countries in question. 

In addition, we obtained information from expert scientists and pest monitoring in the selected countries to obtain up-to-date information on surveillance activities, pest status, and damage. We asked them to send us articles in local languages and to share links for websites where official data on WCR monitoring can be viewed. We also asked them to send us their data on the percentage of infested cropland, an area with economic damage, average yield losses, and average insecticide use (where available).
Category I (Austria, Croatia, Hungary, Italy, Serbia, Slovenia, and Romania)

Because WCR has already spread over the entire territory of Category I countries, there was no need to monitor its distribution at the country level over the last few years. Therefore, we did not find many articles published in the last 10 to 12 years reporting on the national spread, population density, and WCR damage for these countries. 

Research initiatives in the observed countries aimed to identify factors affecting adult and larval populations and damage at field level in Croatia [64,66,72,102,137,139], Hungary [78,162,231], Serbia [66,79,82,104,145,148,224], Slovenia, [53,76] and Romania [52,56,61,68,85,93,97,98,99,140]. 

The articles dealing with population monitoring in Croatia describe monitoring techniques [6,72] rather than population levels and damage. In contrast, the article by Falkner et al. (2019) [234] is based on data collected during WCR monitoring in Austria from 2002 to 2015. In the paper, the authors reported the highest WCR population level in Styria, the province where WCR has occurred since 2002. Based on the collected data, the authors developed a spatial zero-inflation Poisson mixture model (ZIP) to relate WCR counts to climatic conditions and maize proportions and to account for zero inflation and spatial correlation in the counting data. The developed model provides a scientifically sound basis for analyzing the effects of future climate change scenarios and maize rotation restrictions on WCR distribution and abundance. 

Models for efficient WCR management have been developed in Austria [236,238,239], Hungary [233], and Italy [244], showing that crop rotation restrictions can help to reduce WCR spread and abundance regardless of the climate change scenario considered. Therefore, the impact of climate change is limited compared with the impact of crop rotation restriction measures. The developed models show that legislation requiring 100% crop rotation to control WCR seems too strict. Using the meta-models developed in Hungary, one can easily estimate the percentage of maize fields that would promote the increase of a pest population above the threshold. The results can be used by regional or national agricultural policy decision makers, and for integrated pest management. These models are also recommended for Italy with some adaptations [244] and could be useful for decision makers and farmers when planning flexible rotation with arable crops. These plans may allow farmers to plant the maximal maize percentage for a cultivated area, even including some continuous maize fields, which is a move that would also prevent the establishment of an economic damage WCR population. 

Of the seven countries belonging to Category I, official pest monitoring is still carried out in Austria [245] and Serbia [246]. The other five monitored countries (Hungary, Croatia, Slovenia, Italy, and Romania) are no longer conducting official monitoring [247,248,249]. Due to this situation, it was difficult to collect extensive and reliable information on population density and damage. 

Based on the data from WCR monitoring in Austria [245], it is clear that the pest is distributed throughout the maize-growing area in this country. Monitoring activities were carried out throughout Austria’s maize-growing area, except for the Alps. Calculations by Feusthuber et al [236] show a large spatial variability in WCR damage potential on gross markets. They indicated that a large WCR population in a field combined with adverse weather conditions can result in total maize yield loss. As reported by Falkner et al. (2019) [236], there are no official data on damage and yield loss. As the spatial variability in the economic damage potential of maize yield losses corresponds to the regional maize density, the frequency of maize cultivation is already regulated by law in Styria, where the WCR population is high. A maximum maize share of 75% in crop rotations was legally allowed until 2016 and was reduced to 66% in 2017 [239]. In addition to crop rotation, there are chemical insecticides and entomopathogenic nematodes on the Austrian market that are approved for WCR control [228]. Kropf et al. [224] investigated farmers’ behavior concerning individual and collective WCR control measures, and the results suggest that new forms of knowledge transfer are needed to facilitate the proactive implementation of individual and collective WCR control measures before triggering events, such as severe WCR damage. 

In Croatia, official WCR monitoring was discontinued in 2012. The area of Croatia where WCR population density reaches the economic threshold is located in the northern part of Croatia on the border with Hungary and part of Slovenia (Međimurje and Podravina region) next to the River Drava (on approximately 10,000 km^2^ [55]. In this area, maize is grown on approximately 110,000 ha and the share of maize cultivation in agricultural land is over 60% because farmers depend on maize due to intensive livestock farming. To prevent damage, crop rotation is practiced in all fields where the adult WCR population reaches the economic threshold. As a reliable tool enabling farmers to select the most suitable field for continuous management, Kos et al. [139] suggested using Pherocon AM traps (Trece Inc., Adair, OK, USA) between the 29th and 32nd weeks. The estimated WCR adult catch that could cause significant larval infestation is ≥22 adults/trap in the 29th week. Because IPM is mandatory for all Croatian farmers receiving income support through direct payments, crop rotation is also mandatory as one of the elements of IPM. A granulated insecticide (tefluthrin) is approved for larvae control. WCR damage is not officially recorded. 

In Hungary, monitoring activities were stopped because WCR has spread over the whole area. The national survey conducted in the early 2000s showed that root damage occurred in 22.9% of heavily infected continuous maize fields (more than 10 adults/plant/ day). At that time, the greatest damage was measured in Tolna, Baranya, Békés, Bács-Kiskun, and Csongrád counties. In the early 2000s, the pest conquered the best maize-growing areas in Hungary [250]. As in all EU countries, the plant protection regulation in Hungary requires the implementation of the principles of integrated pest management (IPM), as described in Annex III of Directive 2009/128/EC [49], where crop rotation is the first mandatory measure. In the mid-2010s, WCR was deregulated and placed under the general IPM management approach; its population is now managed by the mandatory minimum number of non-host crops in the rotation, and by other components of EU greening. According to recent information collected by plant protection specialists in some regions of Hungary [247], WCR is still an important pest and there are outbreaks of the pest in some regions from year to year, as was the case in 2012 [251] and 2017 [252]. Moreover, Gyeraj et al. [141] reported silking damage caused by adult WCR on sweet corn and the need to determine the economic threshold to justify the area application of insecticides. 

After WCR was successfully eradicated in Europe’s first focus area, which was identified in Italy, around Venice Marco Polo airport in 1998 [63,253], WCR spread to all Italian maize-growing areas. The spread started with new focus sites in Lombardy [254] and Friuli Venezia Giulia [63,255]. After the first WCR crop damage was observed in 2002 in Lombardy [256], new eradication/containment programs were established in Italy [257]. Despite these programs, WCR populations continued their spread, albeit slowly, and increased year by year. Therefore, official national WCR monitoring was carried out until 2012 [255,258,259,260,261,262] when WCR completed its spread to all the major maize-growing areas, including central Italy [261]. From that point, it was clear that economic threshold levels of WCR populations would be reached wherever continuous maize cultivation was the prevalent practice. Therefore, after emergency eradication, large-scale IPM measures were established regularly [262] and were supported by an innovative insurance tool to cover the risk of IPM implementation [244,263]. 

Official monitoring observed that maize crop-damage cases occurred in newly infested areas after 4–5 years from the first beetle captures, with subsequent population stabilization and crop-damage decrease or disappearance [255]. We can hypothesize that this trend was due to the implementation of containment measures with the interruption of continuous maize together with growing farmer awareness of the WCR problem, once farmers had seen WCR damage symptoms directly. The lack of damaged maize fields in 2011 and 2012 might be considered a confirmation of this hypothesis [260,261]. 

Later, during 2016–2017, area-wide (hundreds of hectares) WCR management strategies were assessed in northeast Italy [244]. A strategy based on chemical control (high presence of continuous maize plots with adult treatments and/or seed treatments) was compared to a flexible rotation approach (continuous corn for two or more years, interrupted when WCR populations exceeded the damage threshold of six beetles/Ph AM trap per day averaged over 42 days). WCR beetle levels were found significantly higher in the chemical control scenario than in the flexible rotation scenario, confirming that crop rotation is the most effective strategy for maintaining WCR populations permanently below the damage threshold without insecticide applications, even with a flexible approach. Flexible crop rotation may imply a higher risk of local sporadic damage because it means allowing the beetle’s population to approach its damage threshold; to avoid an economic loss for farmers, this risk has been successfully managed with the introduction of insurance instruments, such as mutual funds [244,263].

In Serbia, organized monitoring collected data on damage to over 140,000 ha of maize until 1999. After 2000 and 2003, WCR population density and the number of damaged maize fields decreased significantly due to the massive application of crop rotations. Due to WCR presence and severe damage in the early phase of the invasion, the proportion of continuous maize in Serbia, once as high as 30%, was reduced to almost 0%. Sivčev et al. studied 794 maize fields from 2002 to 2006, and under the conditions of their study, the ratio of maize to non-maize fields was about 50:50 [145]. Rotation proved to be very effective because 87.8% of the fields lacked an economically harmful WCR population. Although WCR is considered a well-established pest that can be effectively controlled through diversified crop rotation, official WCR monitoring continues. We analyzed data collected between 2013 and 2020 by official monitoring activities available on the official website [264]. In the last eight years, the level of WCR population caught in pheromone traps in Serbia has fluctuated (Table 2), with a peak in 2016 and a significant decrease in the last three years (2018–2020).

For monitoring purposes, pheromone-baited traps were used. Therefore, the maximal daily captures were very high in some fields. Based on the collected data on pheromone-baited traps is not possible to conclude whether or not the WCR population has reached the economic threshold on the monitored field. No official data on yield losses caused by WCR are available but, based on the discussion with experts [246], farmers are paying much attention to crop rotation, thus economic damages are limited to continuous maize fields in the third or fourth year of continuous sowing. As it was reported by Filipović et al., farmers’ education through farmer field schools contributed to a better understanding of why crop rotation plays such an important role in reducing pest damage even though maize has the highest gross margin when compared to soybean and wheat [227].

The status of WCR in Slovenia is the same as in Hungary and Croatia—it is not considered a quarantine pest since it spread over the whole territory of Slovenia in 2009 [248]. According to Razinger [248] crop rotation is the predominant (alternative) control tool for WCR and it is performed on around 80% of maize planted-surface in Slovenia. However, the percentage of monoculture maize production differs a lot among the regions (3% in Pomurje region and almost 1/3 in Gorenjska region). Approximately 30% of maize is treated with 13.3 kg/ha Force (tefluthrin) at sowing against WCR; approximately 60% of maize seed is treated with Sonido (thiacloprid) against wireworms; approximately 10–25% of maize is treated with both, tefluthrin and thiacloprid, at sowing. Often, crop rotation and insecticides (seed coating—Sonido; and granular—Force) are used together. Generally, in Slovenia, economic damage occurs every year on less than 10 ha and yield loss is negligible, i.e., 0–1% every year. To estimate hatching time and adult emergence, a decision-support model based on degree-days is practiced in Slovenia [230]. In addition, the use of entomopathogenic nematodes and pheromone-based mating confusion are under investigation as ecologically acceptable tools for WCR control. 

Until 2010, WCR was a real danger for maize plants in the western part of Romania and the speed of spreading was astounding [123]. After 2010, at the national level, no extensive monitoring research or official data centralization was performed. At that time, it was estimated that the pest was present only in the western half of the country [249]. After 2010, WCR continued to expand slower, probably due to the Carpathian Mountains that prevented the flight of adult forms [93]. Even though adults have not migrated to other parts of the country, they have grown to size in the western populations (where they first appeared) [123]. According to Grozea [249], WCR is still present in Romania, especially in the west of the country (in Timis county), and continues to expand eastward. In 2019, it was also reported in the east of the country (Neamt County). The presence of the species and its extension were probably favored by the cultivation of corn on an appreciable area of approximately 2.5 million hectares and by the practice of monoculture on large areas. The highest population level is by far in the western area (especially in Timis County) where corn is cultivated on 189,000 hectares, the plain area, and there are still farmers who practice monoculture (20%). Studies conducted in this county in the period 2015–2018 showed that the insect population was located at a fairly high level, but still not as in the period 1997–2010 when there were economic losses.
Category II (Germany and France)

The first occurrence of WCR in Germany was recorded in 2007, in Baden-Württemberg, and Bavaria. Beetles were also caught in North Rhine-Westphalia in 2010 and Hesse and Rhineland-Palatinate have also been affected since 2011. WCR is not quarantine pest anymore and WCR monitoring is performed by each of the federal states separately, but data are also sent to JKI, Institute for Plant Protection in Field Crops and Grassland [264]. The federal states Mecklenburg-Pomerania and Schleswig-Holstein are not participating in the WCR monitoring because the species is not a quarantine pest anymore. The traps used in the monitoring are PAL traps (Csalomon^®^ Pheromone Traps) with pheromone dispensers. Most federal states use the products from Trifolio-M, but Csalomon products are also used. According to collected data from monitoring activities [265,266], WCR is spread in six out of ten federal states involved in monitoring in Germany. The federal states that are not infested are Hesse, Lower Saxony, North Rhine-Westfalia, and Thuringia. In the federal states Brandenburg, Rhineland-Palatinate, Saxony-Anhalt, and Saxony, the population level is very low. The regions Baden-Wurttemburg and Bavaria have the highest population level. The number of trapping locations varies markedly between the federal states with heavy infestation such as Baden-Wurttemberg (700 sites in 2019) and Bavaria (243 sites in 2019) and those that have no infestations such as Lower Saxony (90 sites in 2019) or Thuringia (20 sites in 2019). Bayern and Baden-Wurttemberg have had rapid increases in WCR during the last six years. A slight decrease in 2020 is recorded in Bavaria [261]. In contrast to Bavaria in which the beetles came mainly from the southeast spreading northwest and west [267], the main focus of the infestation in Baden-Württemberg is the upper Rhine valley. Here, also the adjacent regions in France are infested. In Saxony, the infestation is likely spreading from the Czech Republic and Poland, and in Rhineland-Palatinate, it is coming from Baden-Wurttemberg. There are about 600,000 hectares of maize in Bavaria, of which about 420,000 hectares are infested [268]. Despite WCR is still spreading and has increased in numbers in recent years, so far, no economic damage has been observed in Germany, even in the most heavily infested federal states. Only occasional damages of plants are visible to the trained eye in parts of the fields. German IPM measure is to have maize at maximum in two of three years on the same fields in the regions where WCR occurs. It can be stated that if farmers grow maize in the same plot only every two or three years, they do not need to fear considerable damage. Even if an infestation is established, no control measures are necessary. Transferred to the situation of Bavarian maize cultivation this implies that of 420,000 ha cultivated maize area, about 30,000 ha can be considered as highly endangered. Southern Bavaria is concerned almost exclusively [269]. 

After the first WCR detections in 2002 in France (near Paris Charles de Gaulle airport) [270], multiple regulatory changes until the deregulation of the species in 2014 occurred. However, official monitoring activities are carried out by Arvalis [270]. WCR is significantly present in only two areas of France—Alsace and Rhône-Alpes. In 2019, for the first time, insects were easily observable (without traps) in certain plots of these two regions. The first damage attributable in part to the WCR was even observed in an area in the Alpine valley in Rhône-Alpes (Grésivaudan valley) where the first captures had taken place 10 years earlier. Moreover, the outbreaks of WCR are multiplying in the great southwest of France and in particular in New Aquitaine. In Alsace, 100% of the 91 pheromone traps revealed the presence of WCR, which confirms once again that the insect is present throughout the region. Given the high intensities of WCR catches in 2018, some parts were monitored using unbaited chromotropic traps (free of sexual pheromone) in 2019. A plot even accounted for 4.7 insects/trap/day, approaching the established economic thresholds of 5–6 beetles/trap/day (i.e., 40 beetles/trap/week). At the end of 2019, the damages were observed on less than 10 hectares (and it was the first time damages were observed in France) [271]. In Alsace and Rhône-Alpes, there are 300,000 hectares of maize (grain and silage), which represents 10% of maize in France (3 million hectares of maize grain + silage). It is estimated [271] that in this area, the WCR population is reaching the economic threshold of less than 0.01 % (i.e., 30 hectares). The insecticides against WCR are not used and crop rotation is recommended. The recommendation to farmers is to consider the establishment of other crops one year out of six when possible, prioritizing their implementation first on plots where the highest levels of catches were observed in 2019.
Category III (Switzerland)

In Switzerland, WCR is under eradication. Due to the crop rotation system, the pest was unable to establish itself in Switzerland [272]. Because the beetle has not spread in Switzerland, WCR is still a quarantine organism in this country and is therefore regulated by phytosanitary law. Any suspected infestation must be reported to the cantonal plant protection service immediately. 

In south Switzerland, near Ticino, WCR has been caught every year since 2000, as the beetles fly in regularly from Italy. In the Canton of Ticino since 2004, the cultivation of maize on maize has been generally prohibited in the entire area. Official monitoring activities have been taking place since 2003 [272]. The WCR situation in Switzerland has been monitored annually using around 150 pheromone traps each year. Traps are set up mainly in the maize growing areas and in places where the beetle was caught the previous year. Until 2019, special attention was paid to the traffic axes and airports. Since 2020, the traps have been distributed in a grid pattern across the entire Swiss maize-growing area because an increasing number of adult corn rootworms have flown in from the surrounding countries. If corn rootworms are caught, it is compulsory to follow a crop rotation restriction (cultivation of maize on maize is prohibited) within 10 kilometers (demarcated area) of the trap location [272]. Probably because the WCR population level is still very low and that crop rotation is regulated, the damages caused by larval feeding have not been reported until now.

The no-control scenario of potential damage cost of WCR infestation in Europe published by Wesseler and Fall foresaw that the economic benefits of WCR control would range between EUR 143 million and EUR 1.739 million. In the analysis, they included 18 countries (at that time 16 EU member countries plus Croatia and Switzerland) [230]. In our research, the current situation in 10 countries (eight EU countries plus Serbia and Switzerland) is analyzed. The analysis shows that WCR has established its population in all countries belonging to category I and, in some regions of countries, belonging to category II. The current situation with WCR leads us to the conclusion that quarantine and agricultural experts in EU countries involved in this analysis correctly approached the WCR problem. The countries involved in this analysis reported that the frequency of continuous maize was from 20 % in Croatia to 35 % in Hungary and Romania. 

Among analyzed countries, there is no country reporting economic damage on a large scale. In addition, the insecticide application for reducing WCR larval damage is a regular practice in Romania and Slovenia. All involved countries are requesting crop rotation as the main measure against WCR. Taking into account that the level of maize dependency of different farms varies, some countries are advocating breaking continuous maize growing in one out of three years (Germany) up to one in six years (France). In Croatia and Serbia, the education of farmers took place to carry out a risk assessment on maize fields destined for repeated sowing [66,139,227]. The need for farmer education is also highlighted by Kropf et al. [228]. Crop rotation is not considered necessary on the fields with a WCR population lower than the economic threshold level established by monitoring. A decision support system for WCR is developed in Slovenia [225]. In Austria, Hungary, and Italy, crop rotation is also advocated, suggesting modulation of its intensity with the use of different models developed by scientists [229,234,240].

## 5. Conclusions

Over the past 12 years, WCR has continued to multiply and spread throughout Europe, infesting new countries and increasing population density. During the same period, scientists in Europe continued their research activities to investigate different aspects of WCR management by implementing a range of approaches to WCR control. The research topics are very diverse, resulting in a considerable amount of new knowledge that can contribute to the development of pest control strategies applicable in EU agricultural systems, which are completely different from agricultural systems in the USA region, where this pest causes the most problems. No major economic damage was observed in any of the countries surveyed. This status confirms that EU countries surveyed are applying appropriate WCR management measures by implementing research findings. Research findings are confirmed by the situation in EU countries and crop rotation has proven itself to be the most effective strategy for maintaining WCR populations permanently below the damage threshold. To maintain forage production on livestock farms at the best level in terms of yield and quality, different models of crop rotation could be applied. The application of different models could enable farmers to grow maize for two or more continuous years, while other crops could be planted according to a flexible scheme when the WCR population increases significantly, whenever the threshold is exceeded. Implementation of available models of crop rotation prevents any application of pesticides following current European legislation (Directive 128/2009/EU). 

Now that the European agricultural system has coexisted with WCR for 35 years, we can conclude that WCR is a potentially serious threat that can be effectively contrasted in all EU countries. Due to intensive research and professional activities leading to specific agricultural practices and the EU’s Common Agricultural Policy, there are crop-rotation rotation-based solutions that can manage this pest properly with negligible impacts on farmers and the environment. In many countries, these solutions are regularly implemented either by policymakers, extension services, or farmers themselves. Therefore, WCR has not become as serious a pest as expected when it was discovered on the majority of European territories.

## Figures and Tables

**Figure 1 insects-12-00195-f001:**
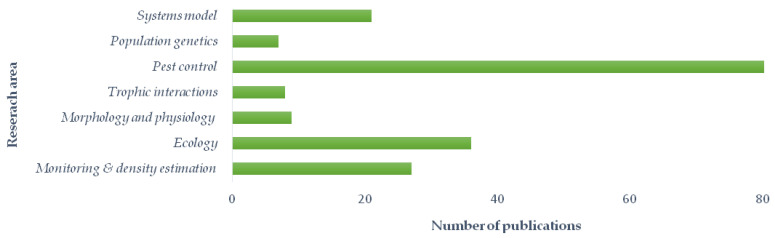
Western corn rootworm (WCR) research area in Europe and published papers accordingly.

**Figure 2 insects-12-00195-f002:**
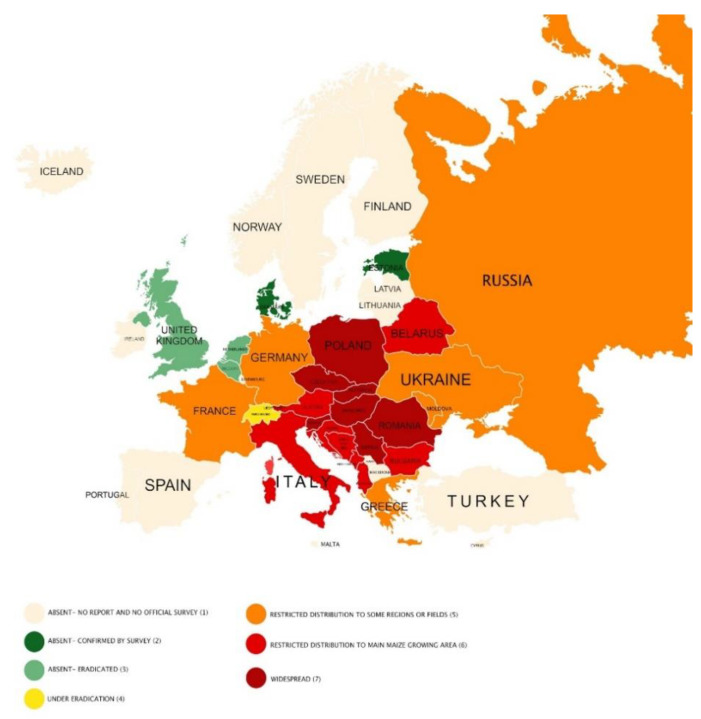
The distribution map of western corn rootworm in Europe—countries are categorized according to pest presence and population level in seven categories (1–7) based on EPPO Global Database [48] and other literature data [53,58,62,69,239], in addition to data reported by Bieńkowski and Orlova-Bienkowskaja, [62] Modič et al. [53], Raileanu and Odobesky, [239] Voineac et al. [63], and Voineac et al. [69].

**Table 1 insects-12-00195-t001:** Overview of the WCR research topics with their main findings and research groups from Europe in the period 2008–2020.

Research Area		Main Findings	Researches’ Group Affiliation Country and Reference	
***Monitoring and density estimation***	**First occurrence and spreading**	the pheromone monitoring is used in two directions: continuous monitoring on territories populated by pest; spreading monitoring—new hearth depistation on a non-populated area	Romania, Slovenia, Poland, Greece, Germany, Russia, Moldova, Italy, Croatia	[52,53,54,55,56,57,58,59,60,61,62]
	**Population level**	the highest abundance was quantified in semi-early and semi-late hybrids	Croatia, Germany, Romania, Ukraine, Slovakia, Poland	[55,63,64,65,66,67,68,69]
	**Monitoring methods/techniques/designs**	traditional monitoring can be effectively used to predict population abundance, and modern monitoring procedures can be used to estimate inter and intra-population variation	Germany, Croatia, Austria, Serbia, Hungary	[6,70,71,72,73,74]
	**Area-wide monitoring**	significant relationship of WCR flight dynamic with the weather and geographical conditions	Slovenia, Romania, Serbia, Hungary	[75,76,77,78]
***Ecology***	**Hosts**	cereals and oil pumpkin plants were not suitable as host plants for larval development while *Miscanthus* sp., are good hosts for WCR	Romania, Poland, Hungary, Austria, Germany, Switzerland, Serbia	[52,79,80,81,82,83,84,85,86,87,88,89,90,91,92]
	**Individual movement**	distance between maize fields and the phenological status of maize influenced inter-field movements	Slovakia, Romania, Hungary	[93,94,95,96]
	**Climatic influence on WCR caches**	physiological limit as a result of climate change might increase the strength of outbreaks at higher latitudes	Romania, Bulgaria, Croatia, Serbia, Spain, Finland	[97,98,99,100,101,102,103,104,105]
	**Soil activities**	plowing and disking had diminishing effects on WCR	Bosnia and Herzegovina, Romania	[106,107,108,109]
	**Attractants**	the most effective in catching WCR were traps with sex attractant	Poland, Croatia, Hungary, Germany	[110,111,112,113,114]
***Morphology and physiology***	**Sexual dimorphism**	sexual dimorphism may be modulated by natural selection	Romania, Germany	[115,116]
	**Wings morphology**	changes in hind wing shape and size are related to identifiable invasion processes	Croatia, Italy, Hungary, Serbia, Austria	[117,118,119,120,121]
	**Enzyme activity**	increase in the esterase activity after pesticide exposure was followed by a significant decrease in AChE	Romania, Poland	[122,123]
***Trophic interactions***	**Influence on root feeding**	larval behaviour respond to root volatiles	Germany, Italy	[124,125,126]
	**Plant signal**	biological control can be improved by manipulating the production of and responsiveness to a plant signal	Switzerland	[127,128]
	**WCR as a disease vector**	WCR may be an important vector of maize fungal diseases	Germany, Croatia, Poland	[129,130,131]
***Pest control***	**Risk assessment**	international research cooperation is the most important key to successfully manage WCR; presented global zones of climatic favourability and invasion risk for the WCR	Spain, Czech Republic, Germany, France	[132,133,134,135]
	**Forecast**	identification of the reproductive potential and longevity of the females of WCR under different rearing conditions	Croatia, Romania, Serbia, Hungary	[65,101,136,137,138,139,140,141,142,143]
	**Crop rotation**	low WCR population in maize fields managed by crop rotation	Serbia, Germany, Romania	[144,145,146,147]
	**Host plant resistance**	significant differences were found in the tolerance levels of the hybrids	Croatia, Germany, Hungary, Italy	[148,149,150,151,152,153,154,155,156]
	**Attract and kill**	host-specific compounds, combined with a CO_2_ source, make attract and kill a feasible management option against WCR	Germany	[157,158,159,160,161]
	**Bt**	proteolytic processing of Bt toxins by WCR midgut juice was examined, no degradation of any of these toxins was observed	Belgium, Hungary, Germany	[162,163]
	**Eradication**	buffer zones large enough to allow eradication are economically unpalatable	UK, Hungary	[164]
	**Chemical control of adults and larvae**	insecticide application led to a significant reduction in the WCR larval density	Netherlands, Poland, Italy, Slovakia, France, Switzerland	[165,166,167,168,169,170,171]
	**Entomopathogenic fungi**	fungal strains significantly influenced the mortality of WCR larvae	Austria, Switzerland, Germany, Slovakia	[172,173,174,175,176,177,178,179]
	**Entomopathogenic nematodes**	nematodes appeared as effective as, or better than standard pesticides at reducing WCR populations	Hungary, Germany, Switzerland, Austria, Serbia, Romania, Italy, Croatia, Belgium, Slovenia	[173,175,178,180,181,182,183,184,185,186,187,188,189,190,191,192,193,194,195,196,197,198]
	**Natural enemies**	natural enemies can be useful elements of a strategic approach to the control of WCR	Switzerland	[199,200,201]
	**Biopesticides**	best results with bioproducts applied to the seed	Hungary, Romania, Slovenia, Germany	[202,203,204,205,206,207]
	**Alternatives and benefits**	annual welfare gain of ca. €190 million from biocontrol of WCR	Italy, France, Italy Spain, Germany, Austria, Romania, Belgium	[208,209,210,211,212,213]
***Population genetics***	**Dispersal**	large European outbreak was expanding by stratified dispersal, involving continuous diffusion and discontinuous long-distance dispersal	France, Italy, UK, Germany, Serbia, Hungary, Austria, Croatia, Slovenia	[214,215,216]
	**Genetic monitoring**	temporal genetic monitoring allowed a deeper understanding of the population genetics of WCR (multiple introductions, admixture, etc.)	Croatia, Serbia, Hungary, Italy	[51,217,218,219,220,221,222]
***Systems model***	**IPM**	management provides a basis for analyzing impacts of climate change and crop rotation on the spread, abundance, and damage of WCR	Austria, Germany, Hungary, Italy, Slovenia, Netherlands, Serbia	[223,224,225,226]
	**Farmers education**	through Farmer Field Schools, farmers were educated on WCR risk assessment	Serbia, Austria	[227,228]
	**Simulations**	development of a mechanistic understanding of the maize-pest system	Netherlands, Hungary, Germany, Sweden, Romania, Austria, Italy	[5,229,230,231,232,233,234,235,236,237,238,239]
	**Remote sensing**	the methodology that identifies WCR larval damage efficiently	Hungary, France, UK	[240,241,242]

**Table 2 insects-12-00195-t002:** The analysis of the WCR monitoring activities in Serbia in the period from 2013 until 2020 (data available at Portal izveštajno prognozne službe zaštite bilja [264]).

	2013	2014	2015	2016	2017	2018	2019	2020
Number of monitored fields	22	24	30	27	29	25	24	25
Percent of fields with the capture	91	87.5	86.7	85.2	86.2	84	75	80
Lowes and highest maximal daily capture	1–186	2–536	1–236	2–250	1–350	1–121	1–442	1–237
Number of fields with maximal daily capture ≥100	3	4	5	12	6	2	1	1

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
