# Peer review of "Western Corn Rootworm (Diabrotica virgifera virgifera LeConte) in Europe: Current Status and Sustainable Pest Management"

_insects, 2021, doi:10.3390/insects12030195_

Round 1

Reviewer 1 Report

Overall, the manuscript draft of Bažok et al. address an important and interesting issue deserving of being reviewed in the rootworm issue for Insects.  The references provided alone, have value.  The English quite good for a group without English as their first language and the review is very complete. 

I do have issues with the structure of the document as is. Overall, it reads more like a bureaucratic report than a scientific review.  There is too much attention to every minor report from every country, but not much synthesis on the contributions these reports made to science.  In addition, the authors seem to think that a funded grant represents a scientific accomplishment (it most certainly does not, in my opinion).  Never before in the refereed literature have I seen a summary of funded grants discussed in chronological order as if they were scientific discoveries.  The stated goals of the manuscript draft were to: “(i) summarize research carried out over the last 12 years in various countries; (ii) describe the EU’s current WCR distribution and population levels and the management strategies implemented; and (iii) assess the potential of structural and flexible crop rotation at area-wide level.”  Why does the reader care about research carried out by various countries?  I for one, care about research accomplishments and new knowledge.

The structure of a review having ‘materials and methods’ and ‘results and discussion’ sections was extremely strange to me.  I suggest that the rotation model in Italy should be removed.  This is a review. Having a mini publication within did not fit the flow of the paper. It seemed to come out of nowhere and I could not put together how everything fit together because it does not.  As far as I am concerned, Lines 220 to 360 can be deleted. At a minimum, they need to be restructured because they do not fit the rest of the paper.  The structure of the manuscript overall was just odd for a review.  It was a summary of what was done in Europe rather than a scientific review.

Regardless of everything above, I do think a modified version would be fine for the rootworm issue of Insects.  It could be that this is exactly what the authors were asked to do.  The writing is generally fine, it is just the structure that seemed odd.  If published, which could be appropriate, I think that more emphasis needs to be put on scientific discoveries than is currently present.

Somewhat minor or specific points that should be addressed

Line 11 – the ‘w’ in western needs to be lowercase here and throughout (including the references cited section!).  This error keeps repeating itself in paper after paper in the refereed literature and is extremely annoying.

Line 15 – delete ‘paper’

Line 38 – what is ‘chromotropic trap beetle captures’? I had to look that up. Regardless, the words ‘beetle’ and ‘trap’ need to be reversed, if kept.  I suggest ‘chromotropic’ be replaced by ‘sticky’.

Line 56 – when did the IWGO first include Diabrotica in its agenda?  It might be good to also note that these meetings were dominated by Diabrotica for a number of years.

Line 66 – consider adding ‘in Europe’ for clarification.

Line 74 – change ‘Muller’ to ‘Miller’; also add a period after ‘et al’

Lines 220 to 230 – text from this section should be deleted or restructured.

Table 1 has some formatting issues with extra digits on the next row. I am not really sure how the table relates to the review, so it too can likely be deleted.

Line 363 – again, why is this not written as a typical review?  187 papers is not a data set.  Decipher and summarize the best of the 187 to tell us what is new and not just what is going on in Europe.  There are 273 references, so how do the 187 relate to the 273? It is not clear.

Table 2 – this is a good summary and should be kept.  I appreciate placing the manuscripts in context and value a number of references that I did not know about.

Line 511 – add a period after ‘et al’

Lines 557-578 – line spacing is different than the other sections.

Lines 698-716 – why the differing indent?

Author Response

REVIEWER 1

Overall, the manuscript draft of Bažok et al. address an important and interesting issue deserving of being reviewed in the rootworm issue for Insects.  The references provided alone, have value.  The English quite good for a group without English as their first language and the review is very complete. 

I do have issues with the structure of the document as is. Overall, it reads more like a bureaucratic report than a scientific review.  There is too much attention to every minor report from every country, but not much synthesis on the contributions these reports made to science.  In addition, the authors seem to think that a funded grant represents a scientific accomplishment (it most certainly does not, in my opinion).  Never before in the refereed literature have I seen a summary of funded grants discussed in chronological order as if they were scientific discoveries.  The stated goals of the manuscript draft were to: “(i) summarize research carried out over the last 12 years in various countries; (ii) describe the EU’s current WCR distribution and population levels and the management strategies implemented; and (iii) assess the potential of structural and flexible crop rotation at area-wide level.”  Why does the reader care about research carried out by various countries?  I for one, care about research accomplishments and new knowledge.

The structure of a review having ‘materials and methods’ and ‘results and discussion’ sections was extremely strange to me.  I suggest that the rotation model in Italy should be removed.  This is a review. Having a mini publication within did not fit the flow of the paper. It seemed to come out of nowhere and I could not put together how everything fit together because it does not.  As far as I am concerned, Lines 220 to 360 can be deleted. At a minimum, they need to be restructured because they do not fit the rest of the paper.  The structure of the manuscript overall was just odd for a review.  It was a summary of what was done in Europe rather than a scientific review.

Response: Rotation model has been removed from the paper.

Regardless of everything above, I do think a modified version would be fine for the rootworm issue of Insects.  It could be that this is exactly what the authors were asked to do.  The writing is generally fine it is just the structure that seemed odd.  If published, which could be appropriate, I think that more emphasis needs to be put on scientific discoveries than is currently present.

Response: The analyse of research results has been added into the text.

Somewhat minor or specific points that should be addressed

Line 11 – the ‘w’ in western needs to be lowercase here and throughout (including the references cited section!).  This error keeps repeating itself in paper after paper in the refereed literature and is extremely annoying.

Response: done

Line 15 – delete ‘paper’

Response: done

Line 38 – what is ‘chromotropic trap beetle captures’? I had to look that up. Regardless, the words ‘beetle’ and ‘trap’ need to be reversed, if kept.  I suggest ‘chromotropic’ be replaced by ‘sticky’.

Response: done

Line 56 – when did the IWGO first include Diabrotica in its agenda?  It might be good to also note that these meetings were dominated by Diabrotica for a number of years.

Response: it was originally described but we have changed the text in order to be more clear:
Since 1995, IWGO, in collaboration with the European Plant Protection Organization (EPPO) and FAO, has organized annual meetings to share new information among scientists on pest distribution and the damage caused. WCR is still on the agenda on regular biannual IWGO meetings.

Line 66 – consider adding ‘in Europe’ for clarification.

Response: done

Line 74 – change ‘Muller’ to ‘Miller’; also add a period after ‘et al’

Response: done

Lines 220 to 230 – text from this section should be deleted or restructured.

Table 1 has some formatting issues with extra digits on the next row. I am not really sure how the table relates to the review, so it too can likely be deleted.

Response: All text and additional materials which were not the result of literature revision process have been removed from manuscript.

Line 363 – again, why is this not written as a typical review?  187 papers is not a data set.  Decipher and summarize the best of the 187 to tell us what is new and not just what is going on in Europe.  There are 273 references, so how do the 187 relate to the 273? It is not clear.

Response: The paragraph under Table 1 (former Table 2) has been rearranged and rewritten, main findings have been added. Table 1 presents an overview of WCR research topics and research groups in Europe from 2008 to 2020. Other references in the literature section (and not in Table 1) were published before this period and are related to projects on WCR, European legislation, research was not conducted by European research groups or was not referenced as scientific papers.

Table 2 – this is a good summary and should be kept.  I appreciate placing the manuscripts in context and value a number of references that I did not know about.

Line 511 – add a period after ‘et al’

Response: done

Lines 557-578 – line spacing is different than the other sections.

Lines 698-716 – why the differing indent?

Response: all technical mistakes have been corrected.

Reviewer 2 Report

Bažok et al. review the Western corn rootworm’s distribution and control measures in Europe, and include empirical data on the effects of rotation on WCR abundance. WCR is an important pest of wide interest globally. The effort to aggregate a substantial amount of information across countries is impressive and commendable. However, the structure of the paper is disjointed. Further, the combination of review and empirical data is awkward, and appears like they were only combined because neither could stand on its own.

Major comments

Overall: I think it is too much to combine the empirical and review components of the paper. They crowd each other out—the appropriate introduction for the review and empirical components is different, as are the methods, results and discussion. As a result of combining them, neither component is done particularly well. I would suggest splitting the paper into two and ensuring sufficient rigor is applied to each. Additionally, I would suggest framing the review around WCR knowledge (like table 2), rather than about specific countries and their monitoring/control activities. The latter sounds more like a government report than a useful advance of WCR science.

Introduction: The introduction does not set up the objectives of the study well. It is at times overly detailed on specific country activities or WCR spread and does not lead into the objectives in a coherent manner. Shortening the introduction to 4-5 paragraphs that introduce the pest, provide a brief history, and elucidate the gaps in knowledge would be far more helpful.

Please make sure to include information on crop rotations and the state of knowledge and gaps related to crop rotations in the introduction if objective III remains in this manuscript. Right now, the empirical research is not well supported by the introduction—don’t we already know that crop rotations are valuable for WCR control? What is the new data adding?

Methods: Methods need more details throughout (See Detailed comments). Use supplementary information if it is too much to include in the main text.

Results: Table 2 is useful. It would be helpful if the authors then organized the information in the results around the research area rather than around the countries.  For example, understanding how climate influences WCR is of broad interest. Detailing what each country is doing in this topic area is of more narrow interest and is more difficult to follow. Synthesize the implications of the studies rather than the locations. Include a table or text on location-specific information in the SI for interested readers.

Conclusion: Please use discussion to refresh the reader on the main points of this study and provide some future research directions to the field. What are the knowledge gaps that future studies should explore?

Detailed comments

Lines 180-184: Please provide the exact search terms, how the papers were screened, and how many were eliminated during the screening process. A flow chart with terms and number of papers would be helpful.

Line 193-194: Based on the EPPO data?

Line 219: How many fields were observed? How frequently were they observed?

Line 234: Selected randomly?

Line 291: How did the authors control for the differences in cultivation techniques and cultivation history? What about year-specific differences in weather or other important determinants of WCR?

Line 444-445: What is the distinction between eradicated and “pest no longer exists”.

Line 549: Confusing language.

Author Response

REVIEWER 2

Bažok et al. review the Western corn rootworm’s distribution and control measures in Europe, and include empirical data on the effects of rotation on WCR abundance. WCR is an important pest of wide interest globally. The effort to aggregate a substantial amount of information across countries is impressive and commendable. However, the structure of the paper is disjointed. Further, the combination of review and empirical data is awkward, and appears like they were only combined because neither could stand on its own.

Major comments

Overall: I think it is too much to combine the empirical and review components of the paper. They crowd each other out—the appropriate introduction for the review and empirical components is different, as are the methods, results and discussion. As a result of combining them, neither component is done particularly well. I would suggest splitting the paper into two and ensuring sufficient rigor is applied to each. Additionally, I would suggest framing the review around WCR knowledge (like table 2), rather than about specific countries and their monitoring/control activities. The latter sounds more like a government report than a useful advance of WCR science.

Response: we have removed the empirical part of the paper. Since the aim of the paper (as it is now) is a combination of the overview of knowledge about WCR and the overview of the current situation, damages and management practices, Chapter 3 is an analysis of the current situation. It may sound like a government report in some parts, but it brings very important information and proves the fact that the implementation of scientific knowledge into agricultural practice has led to a reduction of WCR potential in many countries.

Introduction: The introduction does not set up the objectives of the study well. It is at times overly detailed on specific country activities or WCR spread and does not lead into the objectives in a coherent manner. Shortening the introduction to 4-5 paragraphs that introduce the pest, provide a brief history, and elucidate the gaps in knowledge would be far more helpful.

Response: We have not made a brief introduction. Our understanding is that it is necessary in this form because it establishes the necessity of the paper: In a period we had research activities in Europe focused on the large scientific projects and designed according to the current pest situation. After this period, research activities became more dispersed throughout Europe.

Please make sure to include information on crop rotations and the state of knowledge and gaps related to crop rotations in the introduction if objective III remains in this manuscript. Right now, the empirical research is not well supported by the introduction—don’t we already know that crop rotations are valuable for WCR control? What is the new data adding?

Response: Objective 3 has been removed from manuscript, and the paper is now reorganized.

Methods: Methods need more details throughout (See Detailed comments). Use supplementary information if it is too much to include in the main text.

Response: Part of the manuscript which was based on research have been removed, so we reorganised sections. Details about literature and data collection have been added directly into paragraphs review.

Results: Table 2 is useful. It would be helpful if the authors then organized the information in the results around the research area rather than around the countries.  For example, understanding how climate influences WCR is of broad interest. Detailing what each country is doing in this topic area is of more narrow interest and is more difficult to follow. Synthesize the implications of the studies rather than the locations. Include a table or text on location-specific information in the SI for interested readers.

Response: The main findings of all listed research areas have been added into Table 1 (former Table 2).

Conclusion: Please use discussion to refresh the reader on the main points of this study and provide some future research directions to the field. What are the knowledge gaps that future studies should explore?

 Response: We discussed results of the literature search as a part of the chapter 2 and results regarding the current pest situation and control strategies as a part of the chapter 3. If it is needed, we may extract them in the separate chapter entitled Discussion

Detailed comments

Lines 180-184: Please provide the exact search terms, how the papers were screened, and how many were eliminated during the screening process. A flow chart with terms and number of papers would be helpful.

Response: Details regarding review process have been added. A flow chart has been prepared based on overview in Table 2 and added into manuscript as Figure 1.

Line 193-194: Based on the EPPO data?

Response: It is rephrased: Based on the EPPO Global Database [48]  the selected countries were divided into three categories according the pest distribution status:

Line 219: How many fields were observed? How frequently were they observed?

Response: No longer part of manuscript.

Line 234: Selected randomly?

Response: No longer part of manuscript.

Line 291: How did the authors control for the differences in cultivation techniques and cultivation history? What about year-specific differences in weather or other important determinants of WCR?

Response: No longer part of manuscript.

Line 444-445: What is the distinction between eradicated and “pest no longer exists”.

Response: It is explained in the text: In the United Kingdom and the Netherlands, WCR has been successfully eradicated, and in Belgium the pest no longer exists even though the eradication has not been carried out.

Line 549: Confusing language.

Response: It is rephrased: After WCR was successfully eradicated in Europe’s first focus area, which was identified in Italy, around Venice airport, in 1998 [264,68], WCR spread to all Italian maize-growing areas. The spread started with new focus sites in Lombardy [265] and Friuli Venezia Giulia [267, 68].

Editor:

The editors and reviewers are in general agreement about the strengths and short-comings of the Bažok et al. article. Please address specific comments from each reviewer.

Respones: Done

The editors highlight key areas that need revision below: The editors are in agreement with reviewers that feel the paper as written is very disjointed. The review components of the paper and the empirical data on crop rotation (sections 2.3, 3.3) inserted into the document are incompatible. Please remove the Italy studies (lines 220- 360, 745-824). The Italy empirical data should be published somewhere else. As currently written, the authors are trying to include a publication within a publication. It is well known that crop rotation is a highly effective WCR management tactic. References on crop rotation from Europe could be included in the management section but not unpublished empirical data in a review article.

Response: Iti s removed

The authors review a substantial amount of literature which will be very useful to the scientific community but the editors agree with both reviewers that the organization of results around countries inhibits general conclusions about pest ecology, control, or future research needs. The emphasis needs to be on the collective western corn rootworm knowledge gained over the last 12 years rather than a list of activities conducted by specific countries. A strength of the paper is Table 2 that organizes information around research areas. Please expand the discussion of research and synthesize major findings within research areas across Europe. A suggestion for improving Table 2 would be to use X’s where a country column bisects an area topic row (see attached partial Table as an example.

Response: Done

The organization of current status of the pest by category (I-III) is a very useful way to place pest status and management within the proper context. Consider adjusting some colors in the distribution map to more clearly separate categories

Response: Done

Standardize the reference list to make sure all scientific names are in italics,

Response: done

Adjust the abstracts and conclusions after revising the paper.

Response: done

This review can be a nice contribution to the Special Issue on corn rootworms after revision.

Thank you.

Round 2

Reviewer 1 Report

The manuscript by Bažok et al. is improved and should be accepted for the specific rootworm issue of Insects after revisions. 

Line 90 – delete ‘has’

Line 94 – delete ‘has’

Line 95 – delete ‘has’

Line 101 – add a comma after infrastructure and move ‘no’ before ‘laboratory’

Lines 121-131 – there are more than 260 citations in this review.  Why are the following missing which were funded by the program mentioned here?  They are likely the most important to come from European funding.

  1. Moeser T. Guillemaud. 2009. International cooperation on western corn rootworm ecology research: state‐of‐the‐art and future research. Agricultural and Forest Entomol. 11: 3-7

Spencer, J.L., B.E. Hibbard, J. Moeser, and D.W. Onstad.  2009.  Behavior and ecology of the western corn rootworm (Diabrotica virgifera virgifera LeConte)(Coleoptera: Chrysomelidae).  Agricultural and Forest Entomol. 11: 9-27.

Meinke, L.J., T.W. Sappington, D.W. Onstad, T. Guillemaud, N.J. Miller, J. Komáromi, N. Levay, L. Furlan, J. Kiss, & F. Toth. 2009. Western corn rootworm (Diabrotica virgifera virgifera LeConte) population dynamic. Agricultural and Forest Entomol. 11: 29-46.

Miller, N.J., T. Guillemaud, R. Giordano, B.D. Siegfried, M.E. Gray, L.J. Meinke, T.W. Sappington. 2009. Genes, gene flow and adaptation of Diabrotica virgifera virgifera. Agricultural and Forest Entomol. 11: 47-60.

van Rozen, K. & A. Ester. 2010. Chemical control of Diabrotica virgifera virgifera LeConteJ. Appl. Entomol. 134: 376–384

Figure 1 – ‘Tropic interactions’ should likely be ‘Trophic interactions’

Lines 189 – 258 – when describing specific references, specific references should be cited.

Author Response

REVIEWER 1

The manuscript by Bažok et al. is improved and should be accepted for the specific rootworm issue of Insects after revisions. 

Line 90 – delete ‘has’

Response: Done

Line 94 – delete ‘has’

Response: Done

Line 95 – delete ‘has’

Response: Done

Line 101 – add a comma after infrastructure and move ‘no’ before ‘laboratory’

Response: Done

Lines 121-131 – there are more than 260 citations in this review.  Why are the following missing which were funded by the program mentioned here?  They are likely the most important to come from European funding.

134 Moeser T. Guillemaud. 2009. International cooperation on western corn rootworm ecology research: state‐of‐the‐art and future research. Agricultural and Forest Entomol. 11: 3-7

135 Spencer, J.L., B.E. Hibbard, J. Moeser, and D.W. Onstad.  2009.  Behavior and ecology of the western corn rootworm (Diabrotica virgifera virgifera LeConte)(Coleoptera: Chrysomelidae).  Agricultural and Forest Entomol. 11: 9-27.

221 Meinke, L.J., T.W. Sappington, D.W. Onstad, T. Guillemaud, N.J. Miller, J. Komáromi, N. Levay, L. Furlan, J. Kiss, & F. Toth. 2009. Western corn rootworm (Diabrotica virgifera virgifera LeConte) population dynamic. Agricultural and Forest Entomol. 11: 29-46.

222 Miller, N.J., T. Guillemaud, R. Giordano, B.D. Siegfried, M.E. Gray, L.J. Meinke, T.W. Sappington. 2009. Genes, gene flow and adaptation of Diabrotica virgifera virgifera. Agricultural and Forest Entomol. 11: 47-60.

van Rozen, K. & A. Ester. 2010. Chemical control of Diabrotica virgifera virgifera LeConteJ. Appl. Entomol. 134: 376–384 – was in text

Response: This references have not been included in table because the first author did not have European affiliation. However, we agree that they should be included because at least one of the author and funding were from Europe, and suggested references have been added into table 1 and reference list. 

Figure 1 – ‘Tropic interactions’ should likely be ‘Trophic interactions’

Response: Done

Lines 189 – 258 – when describing specific references, specific references should be cited.

Response: all references related with research area have been cited near group affiliation country, and because of the visibility and readability only one main finding was described. 

Reviewer 2 Report

The authors addressed many of the comments. However, two issues remain related to the organization of the manuscript.

First, the introduction is very long. Lengthy introductions can limit the number of readers that will continue to read the rest of the paper. Perhaps the authors could create a "History of WCR Control in Europe" section at the beginning of Section 2 to provide the detailed chronological information for the interested reader. 

Second, the organization of the review remains around countries rather than around insights. I do not think this is an effective presentation of the information, even with the additional information in now table 1. It remains difficult to synthesize the information or determine useful future research directions. I don't believe the authors have addressed this concern, provided in the initial review, sufficiently. 

Minor:

Line 606-607 suggests there legislation preventing pesticide use for WCR. Then in the next sentence zero pesticide use is described as a "goal". Please check. 

Check grammar and spelling, particularly for newly added sections. 

Author Response

REVIEWER 2

The authors addressed many of the comments. However, two issues remain related to the organization of the manuscript.

First, the introduction is very long. Lengthy introductions can limit the number of readers that will continue to read the rest of the paper. Perhaps the authors could create a "History of WCR Control in Europe" section at the beginning of Section 2 to provide the detailed chronological information for the interested reader. 

Response: We accept reviewers’ suggestion and reorganized introduction part with adding one addition title of paragraph.

Second, the organization of the review remains around countries rather than around insights. I do not think this is an effective presentation of the information, even with the additional information in now table 1. It remains difficult to synthesize the information or determine useful future research directions. I don't believe the authors have addressed this concern, provided in the initial review, sufficiently. 

Response: We understand the reviewer's concern. Taking into account the reviewers' comments, we have already (after first revision) changed the approach to the first goal and discussed the main findings of different research. Therefore the main findings related to the research area have been provided in Table 1 and in the text (pls. see the lines 361-372) On the other side, the second goal of the paper is (ii) to present the current distribution of WCR in the EU and to analyze the current population levels in different European countries, focusing on different management strategies. Keeping this in mind, we focused on the status in different countries and applied management strategies. However we grouped countries according to the pest status. THus we did not change this part.

Minor:

Line 606-607 suggests there legislation preventing pesticide use for WCR. Then in the next sentence zero pesticide use is described as a "goal". Please check. 

Response: The goal was removed from this paragraph, while it is already regulated with legislation.

Check grammar and spelling, particularly for newly added sections. 

Response: done

Round 3

Reviewer 2 Report

This manuscript has a great deal of useful information and citations, most of which will not be recognized by the reader in its current organization around countries. I believe it is a missed opportunity for what could have been a very useful and highly cited contribution.